# Impact of intraoperative transesophageal echocardiogram on changes in surgical management among patients undergoing cardiovascular surgery in Thailand

Naruenart Lomarat[1]*, Chaiyawat Suppasilp[2], Chanpitcha Khumchoei Sidfeldt[1]

**1** Department of Anesthesiology, Faculty of Medicine, Siriraj Hospital, Mahidol University, Bangkok, Thailand, **2** Department of Clinical Epidemiology and Biostatistics, Faculty of Medicine, Ramathibodi Hospital, Mahidol University, Bangkok, Thailand

* naruenart.lom@mahidol.ac.th

## Abstract

Transesophageal echocardiography (TEE) is essential to perioperative cardiac care, providing enhanced cardiac visualization compared to transthoracic echocardiography (TTE), especially in complex cases. While TEE is standard in high-income countries, its utilization in resource-limited settings is not as well-defined. This study aimed to quantify the impact of intraoperative TEE on surgical management at a major tertiary care center in Thailand and to investigate the effects of combining preoperative TTE and TEE on surgical planning. This prospective observational study enrolled 624 adult patients undergoing cardiac surgery from January 2023 to January 2024. All patients received intraoperative TEE, with preoperative assessment conducted via either TTE alone or TTE combined with TEE. The primary outcome was the rate of change in surgical management prompted by new intraoperative TEE findings. Intraoperative TEE findings led to a change in surgical management in 10.58% of all cases (95% CI: 8.28–13.26). The rate of change was higher in patients undergoing preoperative TTE combined with TEE (16.13%) compared to those receiving TTE alone (9.60%); however, after multivariable adjustment, this difference was not statistically significant (adjusted RR 1.18, 95% CI: 0.67–2.09, p = 0.567). The type of surgery was the only independent predictor of management changes, with isolated valve surgery (adjusted RR 2.32, 95% CI: 1.05–5.16) and combined valve with CABG procedures (adjusted RR 3.03, 95% CI: 1.30–7.05) showing the highest likelihood of alteration. Postoperative outcomes, including 30-day mortality and complication rates, were comparable between patients with and without surgical management changes. In this study, intraoperative TEE was associated with changes in surgical decision-making in approximately 10% of cardiac surgeries, suggesting a potential clinical impact, particularly in complex valve-related procedures. The addition of a preoperative TEE, while associated with longer surgical wait times, did not

**Data availability statement:** Due to privacy and ethical restrictions, the dataset cannot be shared publicly because individual patients might be identifiable. Data are available from Chusana Rungjindamai (Please contact via viewfloramu@gmail.com), Research Technical Officer, Department of Anesthesiology, Faculty of Medicine Siriraj Hospital, Mahidol University, Bangkok 10700, Thailand. Access to the data is available upon reasonable request and approval from the Ethics Committee of Siriraj Hospital for researchers who meet the criteria for access to confidential data.

**Funding:** The author(s) received no specific funding for this work.

**Competing interests:** The authors have declared that no competing interests exist.

independently associate with the likelihood of intraoperative changes. These findings underscore the crucial role of intraoperative TEE for real-time assessment and support its selective use in high-complexity cases, while also highlighting logistical challenges within resource-limited healthcare systems.

## Background

The transthoracic echocardiography (TTE) was introduced in the 1950s and has become a cornerstone of cardiac imaging. It offers a noninvasive and accessible approach to assessing cardiac structure and function. Despite its broad clinical utility, TTE diagnostic accuracy can be compromised in specific patient populations due to factors such as obesity, chronic obstructive pulmonary disease, and anatomical abnormalities, including barrel chest deformities [1]. In response to these limitations, the transesophageal echocardiography (TEE) was developed in the 1970s [2]. This technique offers enhanced imaging resolution by positioning the ultrasound probe closer to the heart via the esophagus. This proximity improves visualization of posterior cardiac structures, facilitating more precise diagnoses and enabling real-time intraoperative guidance.

Over the past few decades, TEE has become an essential tool in perioperative cardiac surgery [3]. Major professional societies, such as the American Society of Anesthesiologists and the Society of Cardiovascular Anesthesiologists, endorse its use, particularly during open-heart procedures and coronary artery bypass grafting (CABG) when no contraindications exist. The perioperative roles of TEE include confirming anatomical findings, guiding weaning from cardiopulmonary bypass (CPB), and validating surgical repairs, particularly in valvular interventions [4,5]. Intraoperative TEE impacts surgical decision-making in 4–15% of cases [6–9]. A current systematic review revealed that transesophageal and transthoracic echocardiography are feasible following cardiac surgery and frequently lead to significant changes in clinical management [10].

Despite widespread adoption in high-resource settings, the utilization and availability of TEE in resource-limited countries remain limited due to infrastructural constraints. In Thailand, for instance, cardiac surgery is centralized within a few hospitals, and access to TEE is not uniformly available. This scarcity can result in prolonged waiting periods and potential mismatches between preoperative assessment and intraoperative findings. Significantly, discrepancies between TTE and intraoperative TEE assessments, as well as physiological changes during the interval between imaging and surgery, can alter planned surgical interventions [11,12].

The main objective of this study was to evaluate the impact of intraoperative TEE on surgical management decision-making in the context of a tertiary referral center in Thailand. Furthermore, this study aimed to explore surgical waiting times and the frequency of surgical management changes among those who underwent preoperative TTE alone versus those who underwent preoperative TTE combined with TEE. Lastly, postoperative outcomes were also explored in this resource-limited setting.

## Methods

### Study design and subjects

This prospective observational study was conducted at Siriraj Hospital, following ethical approval from the Siriraj Institutional Review Board (Si 009/2023) and was registered in the Thai Clinical Trials Registry (TCTR20230501001). All patients who underwent cardiac surgery between January 2023 and January 2024 were screened for inclusion. Inclusion criteria were: (1) receipt of cardiac surgery at Siriraj Hospital, regardless of urgency status (elective, urgent, or emergent, defined according to established guidelines [13]), (2) completion of preoperative TTE and/or TEE, and (3) age 18 years or older. Exclusion criteria included complex congenital heart disease, heart transplantation procedures, contraindications for TEE, and failure to undergo intraoperative TEE as a reference investigation for changes in surgical management [5]. The decision to perform intraoperative TEE was made by consensus between the attending cardiac anesthesiologist and the cardiac surgeon immediately prior to surgery. Intraoperative TEE images were interpreted by attending cardiac anesthesiologists with a minimum of two years of experience in cardiac anesthesia and certification by the National Board of Echocardiography (NBE).

### Data collection

Patient characteristics and clinical data—including preoperative echocardiograms (TTE alone or TTE combined with TEE) and the interval between assessment and surgery—were obtained from electronic medical records (EMRs).

The preoperative surgical plan was meticulously documented in the study's case record form using standardized and validated terminology during the time-out period prior to incision. In cases where procedural ambiguity existed (e.g., mitral valve repair ± mitral valve replacement), only the finalized procedure was recorded. For instance, if the plan included both mitral valve repair and replacement as options, but repair was confirmed preoperatively, only 'mitral valve repair' was documented as the intended preoperative procedure.

Intraoperative TEE examinations were conducted during the pre-bypass and post-bypass phases using Philips X7-2t or X8-2t transducers. If a new finding was identified on intraoperative TEE during either the pre-bypass or post-bypass period, the attending cardiac anesthesiologist consulted with the surgical team to evaluate the necessity for modifying the surgical plan.

The primary outcome of this study was the change in the preplanned surgical procedure based on intraoperative TEE findings. Detailed records of surgical changes were maintained, categorized into additions (e.g., incorporating mitral valve repair into a coronary artery bypass grafting procedure) and omissions (e.g., changing from combined mitral and aortic valve replacement to a sole aortic valve replacement). Additionally, any modifications in the surgical plan prompted by factors other than intraoperative TEE findings, such as intraoperative findings, were documented in a similar manner. Postoperative clinical outcomes—including the duration of mechanical ventilation, lengths of ICU and hospital stay, 30-day mortality, and surgical complications —were monitored through September 2025.

### Statistical analysis

Continuous variables were summarized as means with standard deviation (SD) or medians with interquartile range (IQR), as appropriate, while categorical variables were presented as frequencies and percentages. Binomial regression analysis was conducted to identify potential factors associated with changes in surgical management, with the magnitude of association expressed as relative risk (RR). A multivariable model was developed to adjust for potential selection bias and confounding factors. The Wilcoxon rank-sum test was used to compare medians of interval times between groups. Subgroup analyses were performed within the preoperative TTE alone and the preoperative TTE combined with TEE subgroups. Sensitivity analyses were conducted using an alternative outcome definition—surgical management changes attributed to both intraoperative TEE findings and other intraoperative factors. Postoperative clinical outcomes were described and

compared between patients with and without a change in surgical management. All statistical analyses were performed using Stata (StataCorp. 2023. Stata Statistical Software: Release 18. College Station, TX: StataCorp LLC), with a significance level set at 0.05.

## Results

Among 840 patients who underwent cardiac surgery, 624 (74.29%) patients underwent intraoperative TEE and were included in the study (Fig 1). The majority were elderly (66.67%) with a mean age of 62.87 years (SD 14.21). Most procedures were elective (90.87%), with both TTE and TEE predominantly performed in-house. Regarding surgical interventions, over 90% comprised CABG, valve surgery, thoraco-aortic surgery, and their combinations (Table 1). Other surgical procedures (3.85%) included mechanical assist device insertion, myectomy, pulmonary artery thromboendarterectomy, and combined septal defect with valve surgery. Among patients who underwent preoperative TTE alone versus those who received preoperative TTE combined with TEE, significant differences were observed in age, urgency status, surgery type, and location where TTE was performed (p < 0.05). These findings suggest potential selection bias and confounding factors that may have influenced changes in surgical management.

The primary objective of this study was to evaluate the impact of intraoperative TEE in the Thai clinical setting. Findings demonstrated that intraoperative TEE led to changes in surgical management in 10.58% of cases (66/624, 95% CI: 8.28–13.26). Interestingly, patients who underwent preoperative TTE combined with TEE appeared to derive more significant benefit from intraoperative TEE, with a management change of 16.13% (95% CI: 9.32–25.20), compared to 9.60% (95% CI: 7.23–12.43) among those who underwent preoperative TTE alone.

Regarding factors influencing management changes necessitated by intraoperative TEE (Table 2), only the type of surgical intervention was an independent factor in both univariable and multivariable analyses. The adjusted RR was 2.32 (95% CI: 1.05–5.16, p = 0.038) for isolated valve surgery, and 3.03 (95% CI: 1.30–7.05, p = 0.010) for combined CABG with valve surgery; other variables remained non-significant. Details of surgical management changes are provided in Table 3. Preoperative TEE combined with TEE was not significantly associated with changes in surgical management (adjusted RR 1.18, 95% CI: 0.67–2.09, p = 0.567), potentially reflecting selection bias in the decision to perform TEE.

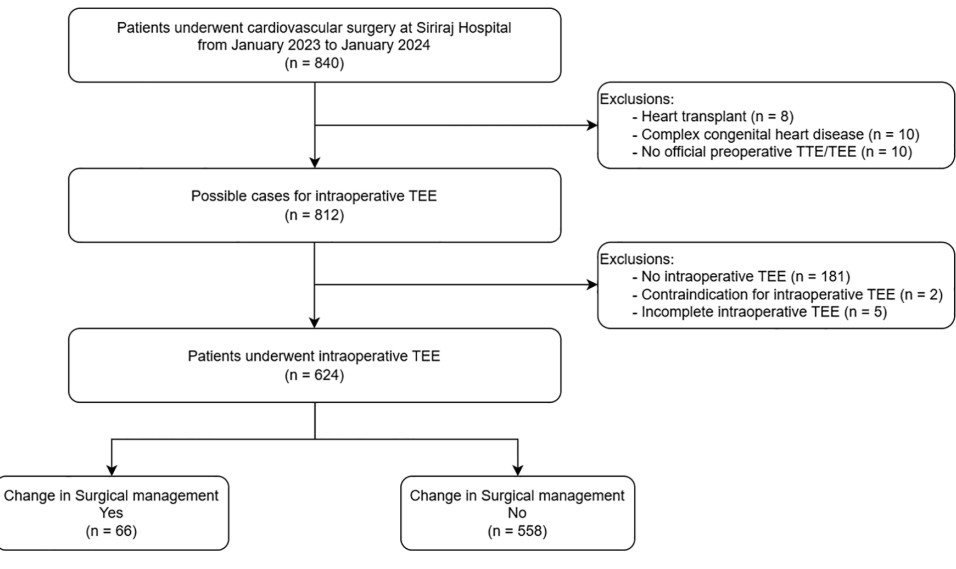

**Fig 1. Study flow.**

**Table 1. Baseline characteristic.**

| Variables | Overall n=624 (100%) | TTE alone n=531 (85%) | TTE combined with TEE n=93 (15%) | p-value |
|---|---|---|---|---|
| Sex | | | | |
| Male | 357 (57.21) | 310 (58.38) | 47 (50.54) | 0.173 |
| Female | 267 (42.79) | 221 (41.62) | 46 (49.46) | |
| Age | | | | |
| Less than 60 years | 208 (33.33) | 160 (30.13) | 48 (51.61) | <0.001 |
| At least 60 years | 416 (66.67) | 371 (69.87) | 45 (48.39) | |
| Status | | | | |
| Elective | 567 (90.87) | 475 (89.45) | 92 (98.92) | 0.003 |
| Urgency/Emergency | 57 (9.13) | 56 (10.55) | 1 (1.08) | |
| Type of Surgery | | | | |
| (1) CABG | 162 (25.96) | 162 (30.51) | 0 (0.00) | <0.001 |
| (2) Valve surgery | 233 (37.34) | 168 (31.64) | 65 (69.89) | |
| (3) Thoracic-aortic surgery | 45 (7.21) | 45 (8.47) | 0 (0.00) | |
| (1) + (2) | 87 (13.94) | 80 (15.07) | 7 (7.53) | |
| (1) + (3) | 7 (1.12) | 7 (1.32) | 0 (0.00) | |
| (2) + (3) | 33 (5.29) | 29 (5.46) | 4 (4.30) | |
| (1) + (2) + (3) | 9 (1.44) | 8 (1.51) | 1 (1.08) | |
| ASD/VSD | 15 (2.40) | 7 (1.32) | 8 (8.60) | |
| Cardiac mass removal | 9 (1.44) | 9 (1.69) | 0 (0) | |
| Other | 24 (3.85) | 16 (3.01) | 8 (8.60) | |
| Location of TTE | | | | |
| In-house | 510 (81.73) | 424 (79.85) | 86 (92.47) | 0.004 |
| Outside | 114 (18.27) | 107 (20.15) | 7 (7.53) | |
| Location of TEE | | | | |
| In-house | 88 (94.62) | NA | 88 (94.62) | NA |
| Outside | 5 (5.38) | | 5 (5.38) | |

Subgroup analysis revealed that changes in surgical management necessitated by intraoperative TEE findings were apparent only in the preoperative TTE alone subgroup (12.98% for valve surgery vs. 5.69% for non-valve surgery), with an RR of 2.28 (95% CI: 1.26–4.12) and a p-value of 0.006. Conversely, no significant differences were found in the preoperative TTE combined with TEE subgroup (15.58% vs. 18.75%), with an RR of 0.83 (95% CI: 0.26–2.61) and a p-value of 0.752.

Regarding waiting times, the median interval from the preoperative TTE to the surgical date was 94.5 days (IQR: 14–211). Among patients who underwent additional TEE, the median interval from the most recent echocardiographic study (either TTE or TEE) to surgery was 92.5 days (IQR: 14–198). As anticipated, patients in the TTE combined with TEE group experienced significantly longer waiting times: 202 days (IQR: 91–299) versus 74 days (IQR: 11–184) from the initial TTE to surgery (p<0.001), and 154 days (IQR: 66–223) versus 74 days (IQR: 11–184) from the last echocardiography to surgery (p<0.001), compared with the TTE alone group. Subgroup analyses between changed versus unchanged managements were presented in S1 Table. S2 Table presents a comparison of waiting times between patients whose surgical management was changed and those whose management remained unchanged. No significant differences in waiting time were observed between the two groups, including in subgroup analyses stratified by the type of preoperative echocardiography.

**Table 2. Crude and adjusted risk ratios for factors associated with intraoperative TEE findings leading to changes in surgical management.**

| Variables | Yes n = 66 (10.58%) | No n = 558 (89.42%) | Crude RR (95%CI) | p-value | Adjusted RR (95%CI) | p-value |
|---|---|---|---|---|---|---|
| Preoperative TEE | | | | | | |
| Yes (TTE + TEE) | 15 (16.13) | 78 (83.87) | 1.68 (0.99, 2.86) | 0.056 | 1.18 (0.67, 2.09) | 0.567 |
| No (TTE alone) | 51 (9.60) | 480 (90.40) | 1 | | 1 | |
| Sex | | | | | | |
| Male | 39 (10.92) | 318 (89.08) | 1.02 (0.78, 1.33) | 0.896 | 1.13 (0.70, 1.83) | 0.611 |
| Female | 27 (10.11) | 240 (89.89) | 1 | | 1 | |
| Age | | | | | | |
| Less than 60 years | 27 (12.98) | 181 (87.02) | 1 | 0.167 | 1 | 0.382 |
| At least 60 years | 39 (9.38) | 377 (90.62) | 0.72 (0.46, 1.15) | | 0.80 (0.48, 1.33) | |
| Status | | | | | | |
| Elective | 63 (11.11) | 504 (88.89) | 1 | 0.193 | 1 | 0.445 |
| Urgency/Emergency | 3 (5.26) | 54 (94.74) | 0.47 (0.15, 1.46) | | 0.63 (0.19, 2.07) | |
| Type of Surgery | | | | | | |
| (1) CABG | 8 (4.94) | 154 (95.06) | 1 | – | 1 | – |
| (2) Valve surgery | 31 (13.30) | 202 (86.70) | 2.69 (1.27, 5.71) | 0.010* | 2.32 (1.05, 5.16) | 0.038* |
| (3) Thoracic-aortic surgery | 2 (4.44) | 43 (95.56) | 0.90 (0.20, 4.09) | 0.891 | 1.02 (0.21, 5.06) | 0.980 |
| (1) + (2) | 13 (14.94) | 74 (85.06) | 3.03 (1.30, 7.02) | 0.010* | 3.03 (1.30, 7.05) | 0.010* |
| (1) + (3) | 1 (14.29) | 6 (85.71) | 2.89 (0.42, 20.06) | 0.282 | 2.63 (0.36, 19.24) | 0.340 |
| (2) + (3) | 4 (12.12) | 29 (87.88) | 2.45 (0.78, 7.68) | 0.123 | 2.31 (0.73, 7.35) | 0.157 |
| (1) + (2) + (3) | 1 (11.11) | 8 (88.89) | 2.25 (0.31, 16.09) | 0.419 | 2.14 (0.29, 15.89) | 0.455 |
| ASD/VSD | 2 (13.33) | 13 (86.67) | 2.70 (0.63, 11.58) | 0.181 | 2.13 (0.45, 10.03) | 0.337 |
| Cardiac mass removal | 0 (0.00) | 9 (100.00) | NA | NA | NA | NA |
| Other | 4 (16.67) | 20 (83.33) | 3.38 (1.10, 10.35) | 0.033* | 2.96 (0.93, 9.41) | 0.066 |
| Location of TTE | | | | | | |
| In house | 58 (11.37) | 452 (88.63) | 1 | 0.183 | 1 | 0.295 |
| Outside | 8 (7.02) | 106 (92.98) | 0.62 (0.30, 1.26) | | 0.68 (0.33, 1.39) | |

**Table 3. Details of surgical management changes.**

| Operations | Change due to only intraoperative TEE | | | Change due to any causes (TEE or others) | | |
|---|---|---|---|---|---|---|
| | Total n = 66 (10.58%) | Addition n = 45 (68.18%) | Omission n = 21 (31.82%) | Total n = 165 (26.48%) | Addition n = 133 (80.61%) | Omission n = 32 (19.39%) |
| (1) CABG | 8 (4.94) | 8 (100.00) | 0 (0.00) | 8 (4.94) | 8 (100.00) | 0 (0.00) |
| (2) Valve surgery | 31 (13.30) | 20 (64.52) | 11 (35.48) | 76 (32.62) | 63 (82.89) | 13 (17.11) |
| (3) Thoracic-aortic surgery | 2 (4.44) | 2 (100.00) | 0 (0.00) | 19 (42.22) | 17 (89.47) | 2 (10.53) |
| (1) + (2) | 13 (14.94) | 6 (46.15) | 7 (53.85) | 23 (26.44) | 16 (69.57) | 7 (30.43) |
| (1) + (3) | 1 (14.29) | 1 (100.00) | 0 (0.00) | 4 (57.14) | 3 (75.00) | 1 (25.00) |
| (2) + (3) | 4 (12.12) | 2 (50.00) | 2 (50.00) | 19 (57.58) | 14 (73.68) | 5 (26.32) |
| (1) + (2) + (3) | 1 (11.11) | 1 (100.00) | 0 (0.00) | 4 (44.44) | 3 (75.00) | 1 (25.00) |
| ASD/VSD | 2 (13.33) | 2 (100.00) | 0 (0.00) | 2 (13.33) | 2 (100.00) | 0 (0.00) |
| Cardiac mass removal | 0 (0.00) | 0 (0.00) | 0 (0.00) | 3 (33.33) | 3 (100.00) | 0 (0.00) |
| Other | 4 (16.67) | 3 (75.00) | 1 (25.00) | 8 (33.33) | 4 (50.00) | 4 (50.00) |

The sensitivity analysis evaluated the outcome of surgical management changes, attributable to intraoperative TEE findings and other intraoperative factors. The overall rate of management changes was 26.60% (95% CI: 23.27–30.26). The associated risk factor analysis yielded similar results, but most surgical procedures were associated with a higher likelihood of management changes compared to isolated CABG (S3 Table). The remaining results, including those related to waiting times, were consistent with the primary findings (S4 and S5 Tables).

In terms of postoperative clinical outcomes (Table 4), the median duration of mechanical ventilation, length of ICU stays, and length of hospital stay were 21 hours (IQR: 15.5–32.25), 1 day (IQR: 1–3), and 9 days, respectively. The 30-day mortality was 5.77% (36/624, 95% CI: 4.07–7.90). The overall complication rate was 38.78% (242/624, 95% CI: 34.94–42.73), with atrial fibrillation (AF) being the most common complication at 14.58%. There was no significant difference in complication rates between patients who did and did not experience changes in surgical management.

## Discussion

In this prospective cohort study involving 624 cardiac surgery patients, intraoperative TEE led to changes in surgical management in approximately 10.58% (95% CI: 8.28–13.26) of cases, aligning with previously published studies highlighting intraoperative echocardiography's diagnostic value and utility in cardiac surgery, which range from 4% to 15% [6–9]. The patient cohort predominantly consisted of elderly individuals undergoing elective surgeries, with CABG, valve surgery, and thoraco-aortic surgery representing the majority of interventions. The selective use of preoperative TEE, primarily in younger patients and those undergoing valve or congenital cardiac repairs, suggests intentional patient selection based on perceived complexity or the necessity of detailed echocardiographic assessment, potentially introducing selection bias and confounding effects.

There were two patient groups: those who underwent preoperative TTE alone and those who received preoperative TTE combined with TEE. This cohort performed preoperative TEE in only 15% of all cardiac surgery cases (n = 93/624),

**Table 4. Postoperative clinical outcomes in patients with and without a change in surgical management due to intraoperative TEE; median (Q1, Q3).**

| Outcome | Total (n = 624) | Change in Surgical management (n = 66) | No Change in Surgical management (n = 558) | p-value |
|---|---|---|---|---|
| Duration of mechanical ventilation (hour) | 21 (15.5, 32.25) | 21 (14, 25) | 21 (16, 34) | 0.199 |
| Length of ICU stays (day) | 1 (1, 3) | 1 (1, 3) | 1 (1, 3) | 0.664 |
| Length of hospital stays (day) | 9 (6, 13.5) | 10 (8, 15) | 9 (6, 13) | 0.070 |
| 30-day mortality; n (%) | 36 (5.77) | 5 (7.58) | 31 (5.56) | 0.573 |
| Complications; n (%) | 242 (38.78) | 25 (37.88) | 217 (38.89) | 0.873 |
| Atrial fibrillation | 91 (14.58) | 11 (16.67) | 80 (14.34) | 0.612 |
| Other arrhythmia | 66 (10.58) | 11 (16.67) | 55 (9.86) | 0.089 |
| Post pericardiotomy syndrome | 3 (3.00) | 0 (0.00) | 3 (0.54) | 1.000 |
| Permanent pacemaker implantation | 20 (3.21) | 4 (6.06) | 16 (2.87) | 0.151 |
| Reoperation (bleeding, tamponade, etc.) | 33 (5.29) | 2 (3.03) | 31 (5.56) | 0.564 |
| Reintubation | 22 (3.53) | 2 (3.03) | 20 (3.58) | 1.000 |
| Pneumonia/ARDS | 49 (7.85) | 3 (4.55) | 46 (8.24) | 0.291 |
| Sepsis | 35 (5.61) | 4 (6.06) | 31 (5.56) | 0.779 |
| Neurological deficit (stroke, etc.) | 38 (6.09) | 3 (4.55) | 35 (6.27) | 0.787 |
| Acute kidney injury (AKI, CRRT) | 11 (1.76) | 0 (0.00) | 11 (1.97) | 0.617 |
| GI complication (bleeding, ischemia, etc.) | 9 (1.44) | 0 (0.00) | 9 (1.44) | 0.608 |
| Mechanical support (ECMO, IABP) | 23 (3.69) | 1 (1.52) | 22 (3.94) | 0.497 |

reflecting the limited resources for conducting an appropriate preoperative evaluation. But, interestingly, the preoperative TTE combined with TEE group demonstrated a higher proportion of management change at 16.13% (95% CI: 9.32–25.20), nearly double that of the preoperative TTE alone group, which was 9.60% (95% CI: 7.23–12.43). This difference may be explained by the selection of more complex cases, as previously noted, and the significantly longer waiting time observed in the preoperative TTE combined with TEE group (median = 202 days, IQR: 91–299), compared to the preoperative TTE alone group (median 74 days, IQR: 11–184). While clinical status could potentially shift during such intervals, the multivariable results showed that combining preoperative TTE combined with TEE did not significantly increase the likelihood of a change in surgical management (adjusted RR 1.18, 95% CI: 0.67–2.09, p = 0.567). The only statistically significant factor was the type of cardiac surgery.

This surgical management change was predominantly observed in valve surgery or combined interventions rather than isolated CABG or thoraco-aortic procedures. This disparity highlights that intraoperative TEE holds greater clinical relevance in cases with complex cardiac anatomy, where real-time echocardiographic assessment can significantly influence surgical decisions. The minimal impact of intraoperative TEE observed among isolated CABG cases aligns with previous studies [14,15], which indicates a limited incremental benefit of routine intraoperative TEE use during uncomplicated revascularization procedures. Nevertheless, intraoperative TEE assessment may offer excellent clinical value in moderate-to-high-risk CABG populations [14,16]. However, the present study did not stratify patients undergoing isolated CABG by risk categories, precluding further analysis of the differential impacts within these subgroups. The adjusted RR for the surgical management change was particularly pronounced in valve surgery, both isolated (RR 2.32, 95% CI: 1.05–5.16) and combined with CABG (RR 3.03, 95% CI: 1.30–7.05). These findings confirm that valve-related interventions involve more significant diagnostic uncertainty, in which intraoperative imaging substantially affects surgical planning and execution [17,18].

This study assessed two waiting time intervals: the interval from preoperative TTE to the surgical date, and the interval from the most recent echocardiogram (TTE in the TTE alone group, and TEE in the TTE combined with TEE group) to the surgical date. As reported in the results section, both intervals were significantly longer in the preoperative TTE combined with TEE group (p < 0.001). Subgroup analyses presented in S1 Table also demonstrated longer waiting times across most of the subgroups. However, among patients who experienced a change in surgical management, the interval from the most recent echocardiogram to surgery did not differ significantly, neither overall (p = 0.171) nor in the elective surgery subgroup (p = 0.372). While disease status may potentially evolve during the preoperative period, particularly in complex cases, this data showed no significant difference in waiting times between patients with and without management changes in either the preoperative TTE alone or preoperative TTE combined with TEE subgroups (S2 Table). These findings suggest that, in this study, waiting time was not independently associated with surgical management changes, though it remains a known risk factor for morbidity and mortality after cardiac surgery [19,20]. Sensitivity analyses were conducted using an alternative outcome definition that included surgical management changes attributable to any causes, including both intraoperative TEE findings and other intraoperative factors, as shown in S3–S5 Tables. Beyond preoperative information, intraoperative management alterations were observed in approximately 26.60% of patients, underscoring the dynamic and multifactorial nature of cardiac surgical decision-making and the need for flexible intraoperative strategies. Consistent with the main findings, multivariable analysis revealed that preoperative TEE was associated with an increased, though not statistically significant, risk of management change (adjusted RR 1.27, 95% CI: 0.94–1.72, p = 0.116). The only independent predictor of management change was the type of surgery, with all categories demonstrating a significantly higher risk compared to isolated CABG (p < 0.05). Waiting time intervals remained consistent with the primary analysis. These sensitivity analyses further support the clinical relevance of perioperative TEE.

Although postoperative outcomes—including duration of mechanical ventilation, ICU and hospital stay, 30-day mortality, and surgical complications—were comparable between patients with and without intraoperative management changes, this does not diminish the importance of TEE. Intraoperative TEE facilitated the identification of unanticipated findings

in patients with complex or severe pathology. While postoperative outcomes were similar between the "change" and "no-change" groups, the observational study design and the universal use of intraoperative TEE preclude a direct assessment of whether these modifications prevented adverse outcomes. Direct comparison with patients of similar risk who did not undergo TEE would be both unethical and clinically inappropriate. Alternatively, these findings may indicate that while TEE effectively detects anatomical and functional abnormalities that prompt procedural modification, the study was not designed to establish a direct causal relationship between these modifications and a reduction in postoperative risk, or that such benefits are not fully reflected in short-term postoperative metrics.

### Strengths, limitations, and future research directions

This study's strengths include its prospective data collection, the application of multivariable regression to control for potential confounders, and comprehensive statistical analyses from multiple perspectives. Nonetheless, a key limitation is the potential for selection bias in assigning patients to preoperative TTE combined with TEE, which is likely influenced by surgical complexity or specific preoperative echocardiographic indications. Furthermore, as all patients received intraoperative TEE, the lack of a control group precludes a direct assessment of whether the absence of TEE-guided modifications would have resulted in worse outcomes. Moreover, the generalizability of the findings may be constrained by the single-center setting and regional practice variations.

Echocardiographic characteristics—both preoperative and perioperative—may differentially influence surgical decision-making and warrant further investigation, particularly within specific surgical contexts such as valve procedures. Future research should aim to identify additional predictors of management changes and assess their broader clinical implications, including surgical duration, timeliness during the operation, and economic evaluation, especially in limited-resource settings.

### Conclusion

Intraoperative TEE altered surgical management in over 10% of cardiac surgeries, primarily in complex valvular and combined procedures, confirming its value for real-time assessment. While postoperative outcomes were comparable between patients with and without TEE-guided changes, the observational nature of this study does not permit a direct conclusion regarding the prevention of adverse events. Preoperative TEE was not an independent predictor of these changes but was associated with longer surgical waiting times, highlighting logistical challenges. Our findings support the use of intraoperative TEE to enhance surgical precision, particularly in high-complexity cases.

### Supporting information

**S1 Table. Waiting time intervals between TTE alone and TTE combined with TEE groups by surgical management change status due to intraoperative TEE.**
(PDF)

**S2 Table. Waiting time intervals between surgical management change statuses due to intraoperative TEE, stratified by preoperative echocardiography subgroups.**
(PDF)

**S3 Table. Sensitivity analyses for changes in surgical management due to any causes: Crude and adjusted RR.**
(PDF)

**S4 Table. Waiting time intervals between TTE alone and TTE combined with TEE groups by surgical management changes due to any causes.**
(PDF)

**S5 Table. Waiting time intervals between surgical management change statuses due to any cause, stratified by preoperative echocardiography subgroups.**
(PDF)

## Acknowledgments

I would like to express my sincere gratitude to Dr. Orawan Supapueng, Division of Clinical Epidemiology Unit, Department of Research, Faculty of Medicine Siriraj Hospital, Mahidol University, for her expert guidance and support in the statistical analysis of this study.

## Author contributions

**Conceptualization:** Naruenart Lomarat.

**Data curation:** Naruenart Lomarat, Chanpitcha Khumchoei Sidfeldt.

**Formal analysis:** Naruenart Lomarat, Chaiyawat Suppasilp.

**Methodology:** Naruenart Lomarat, Chaiyawat Suppasilp.

**Project administration:** Naruenart Lomarat, Chanpitcha Khumchoei Sidfeldt.

**Writing – original draft:** Naruenart Lomarat, Chaiyawat Suppasilp.

**Writing – review & editing:** Naruenart Lomarat, Chaiyawat Suppasilp.

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
