## [Decision Letter · Decision Letter 0]

4 Sep 2025

Dear Dr. Lomarat,

Thank you for submitting your manuscript to PLOS ONE. After careful consideration, we feel that it has merit but does not fully meet PLOS ONE’s publication criteria as it currently stands. Therefore, we invite you to submit a revised version of the manuscript that addresses the points raised during the review process.

We look forward to receiving your revised manuscript.

Kind regards,

Wei Wu

Academic Editor

PLOS ONE

**Journal Requirements:**

1. When submitting your revision, we need you to address these additional requirements. Please ensure that your manuscript meets PLOS ONE's style requirements, including those for file naming. The PLOS ONE style templates can be found at https://journals.plos.org/plosone/s/file?id=wjVg/PLOSOne_formatting_sample_main_body.pdf and https://journals.plos.org/plosone/s/file?id=ba62/PLOSOne_formatting_sample_title_authors_affiliations.pdf 2. If the reviewer comments include a recommendation to cite specific previously published works, please review and evaluate these publications to determine whether they are relevant and should be cited. There is no requirement to cite these works unless the editor has indicated otherwise. 

**Additional Editor Comments:**

1, Although the manuscript suggests that intraoperative TEE was performed in all included cases, this point is not consistently or clearly stated. Given that the primary outcome relates directly to TEE’s influence on surgical planning, it is essential to clearly document the TEE rate and whether it was applied uniformly to all patients. In addition, the manuscript would benefit from more clarity on patient selection, flow, and subgroup structure. For example, it remains somewhat confusing how the subgroups (e.g., TTE-only, TEE-only, or both) were formed, and whether TEE indications were predefined or at the discretion of the care team.

2, The lack of postoperative outcome analysis limits the ability to fully assess the clinical impact of TEE-guided surgical plan changes. While this may have been outside the original scope, the inclusion of even limited outcome data (e.g., 30-day mortality, complication rates, or length of stay) would substantially strengthen the conclusions and may be feasible from prospective dataset.

Reviewers' comments:

**Comments to the Author**

1. Is the manuscript technically sound, and do the data support the conclusions?

Reviewer #1: Partly

2. Has the statistical analysis been performed appropriately and rigorously?

Reviewer #1: No

3. Have the authors made all data underlying the findings in their manuscript fully available?

Reviewer #1: Yes

4. Is the manuscript presented in an intelligible fashion and written in standard English?

Reviewer #1: Yes

**Reviewer #1:** # Summary

1. These investigators conducted a prospective observational study to investigate the impact of intraoperative TEE on surgical planning (primary outcome) and wait times (secondary outcome), among 624 cardiac surgical patients undergoing surgery the calendar year of 2023. They found intraoperative TEE altered the surgical plan in 10% of cases, but did not study clinical outcomes.

# Overall Assessment

1. This is important work on the impact of TEE in a resource-limited situation. However, the absence of investigating TEE's impact on postsurgical outcomes limits the impact and makes this study largely descriptive.

# Major Issues / Questions

1. what was the overall rate of intraoperative TEE? I see the rate of preoperative TEE was only 15% but I don't see intraoperative TEE reported. If intraoperative TEE was performed in only 15% of cases, there is a major selection bias and residual confounding is likely present. At a minimum matching would be necessary and could consider 1:2 or even 1:3 matching if TEE rate is truly only 15%.

2. While a change in surgical plan is important, it would be stronger if there were an associated postoperative outcomes analysis (e.g. lower rates of reoperation, stroke, complications, death, etc. with TEE)?

# Minor Issues

1. The terminology needs to be consistent. Decide on "perioperative" or "intraoperative" TEE and keep it consistent throughout.

2. Consider limiting the analysis to intraoperative TEE (instead of adding TTE and the wait times). It starts to become confusing to the reader to interpret.

**Do you want your identity to be public for this peer review?** For information about this choice, including consent withdrawal, please see our Privacy Policy

Reviewer #1: No

---

## [Author Response · Author response to Decision Letter 1]

2 Nov 2025

Dear Wei Wu,

I hope this message finds you well. We are pleased to submit the revised version of our manuscript entitled “Impact of Perioperative Transesophageal Echocardiogram and Preoperative Echocardiography Waiting Time on Surgical Management in Thailand” (Manuscript Number: PONE-D-25-35063), following the helpful and insightful comments provided by you and the reviewers. We would like to express our sincere gratitude for the constructive feedback, which has significantly improved the quality of our work.

After thorough discussion with our team, we have categorized the comments into items, as outlined below.

Editor’s comments:

1. Although the manuscript suggests that intraoperative TEE was performed in all included cases, this point is not consistently or clearly stated. Given that the primary outcome relates directly to TEE’s influence on surgical planning, it is essential to clearly document the TEE rate and whether it was applied uniformly to all patients. In addition, the manuscript would benefit from more clarity on patient selection, flow, and subgroup structure. For example, it remains somewhat confusing how the subgroups (e.g., TTE-only, TEE-only, or both) were formed, and whether TEE indications were predefined or at the discretion of the care team.

Response: We have revised the manuscript to state explicitly that intraoperative TEE was performed on a cohort representing 74.29% of all cardiac surgeries during the study period; this is now clarified in the first paragraph of the Results section and in the inclusion/exclusion criteria. We have also refined the Methods section under "Study design and subjects" to define the indications for intraoperative TEE, noting that the decision was based on a joint agreement between the attending cardiac anesthesiologist and surgeon.

To address the confusion regarding patient groups, we have clarified that the "TTE-only" and "TTE+TEE" designations refer to preoperative imaging assessments and were analyzed as factors of interest, not as prospectively formed study groups. The revised Figure 1 (Study Flow Diagram) now more clearly illustrates patient selection and flow. Finally, we have standardized the terminology to "intraoperative TEE" throughout the manuscript for consistency.

2. The lack of postoperative outcome analysis limits the ability to fully assess the clinical impact of TEE-guided surgical plan changes. While this may have been outside the original scope, the inclusion of even limited outcome data (e.g., 30-day mortality, complication rates, or length of stay) would substantially strengthen the conclusions and may be feasible from prospective dataset.

Response: We appreciate and agree completely with this valuable suggestion. We have now incorporated a comprehensive analysis of postoperative outcomes through September 2025. This new dataset includes the duration of mechanical ventilation, length of ICU and hospital stays, 30-day mortality, and the incidence of major complications (e.g., atrial fibrillation, reoperation, stroke, acute kidney injury). The Results and Discussion sections have been substantially revised to integrate these findings, providing a more robust assessment of the clinical implications of TEE-guided management changes.

Reviewer’s comments

1. Is the manuscript technically sound, and do the data support the conclusions?

Reviewer #1: Partly

Response: With the addition of the comprehensive postoperative outcomes analysis and the clarifications made in response to all editor and reviewer comments, we believe the manuscript is now technically sound and that the conclusions are more robustly supported by the data.

2. Has the statistical analysis been performed appropriately and rigorously?

Reviewer #1: No

Response: We have re-examined our methodology and confirm that standard, appropriate statistical methods were applied, consistent with the study's prospective observational design and data characteristics, as detailed in the Statistical Analysis section. Based on our best knowledge, we did it appropriately but would be grateful for any specific suggestions for further improvement.

3. Have the authors made all data underlying the findings in their manuscript fully available?

Reviewer #1: Yes

4. Is the manuscript presented in an intelligible fashion and written in standard English?

Reviewer #1: Yes

5. Review Comments to the Author

Reviewer #1: # Summary

1. These investigators conducted a prospective observational study to investigate the impact of intraoperative TEE on surgical planning (primary outcome) and wait times (secondary outcome), among 624 cardiac surgical patients undergoing surgery the calendar year of 2023. They found intraoperative TEE altered the surgical plan in 10% of cases, but did not study clinical outcomes.

# Overall Assessment

1. This is important work on the impact of TEE in a resource-limited situation. However, the absence of investigating TEE's impact on postsurgical outcomes limits the impact and makes this study largely descriptive.

Response: As detailed in our response to the editor (Comment #2), we have now conducted and integrated a full analysis of postoperative clinical outcomes. This addition moves the study beyond a descriptive focus and provides crucial data on the clinical impact of TEE-guided decisions, significantly enhancing the manuscript's contribution.

# Major Issues / Questions

1. what was the overall rate of intraoperative TEE? I see the rate of preoperative TEE was only 15% but I don't see intraoperative TEE reported. If intraoperative TEE was performed in only 15% of cases, there is a major selection bias and residual confounding is likely present. At a minimum matching would be necessary and could consider 1:2 or even 1:3 matching if TEE rate is truly only 15%.

Response: We apologize for the lack of clarity. The 15% rate referred to preoperative TEE. We have now clarified in the Methods and Results sections, as well as in the revised Figure 1, that intraoperative TEE was performed in 624 patients, constituting 74.29% of all cardiac surgeries during the study period. This high rate of utilization strengthens the validity of our findings regarding its impact.

2. While a change in surgical plan is important, it would be stronger if there were an associated postoperative outcomes analysis (e.g. lower rates of reoperation, stroke, complications, death, etc. with TEE)?

Response: As stated in response to editor’s comments and overall assessment of the reviewer’s comments.

# Minor Issues

1. The terminology needs to be consistent. Decide on "perioperative" or "intraoperative" TEE and keep it consistent throughout.

Response: We thank the reviewer for pointing this out. The manuscript has been revised to use the term "intraoperative TEE" consistently throughout.

2. Consider limiting the analysis to intraoperative TEE (instead of adding TTE and the wait times). It starts to become confusing to the reader to interpret.

Response: To streamline the main text, we have moved the detailed results and tables related to preoperative imaging and waiting times into the Supplementary Information. The primary manuscript now focuses more sharply on the impact of intraoperative TEE, while still retaining the waiting time discussion to provide essential context regarding resource limitations in our setting.

6. PLOS authors have the option to publish the peer review history of their article (what does this mean?). If published, this will include your full peer review and any attached files.

Reviewer #1: No

We believe that these revisions have strengthened the manuscript, and we hope that the revised version meets the expectations of both you and the reviewers. We look forward to your feedback and hope for a favorable consideration of our submission for publication in PLOS ONE.

Thank you again for your time and consideration.

Sincerely,

Naruenart Lomarat

---

## [Decision Letter · Decision Letter 1]

19 Nov 2025

Dear Dr. Lomarat,

Thank you for submitting your manuscript to PLOS ONE. After careful consideration, we feel that it has merit but does not fully meet PLOS ONE’s publication criteria as it currently stands. Therefore, we invite you to submit a revised version of the manuscript that addresses the points raised during the review process.

We look forward to receiving your revised manuscript.

Kind regards,

Wei Wu

Academic Editor

PLOS ONE

Journal Requirements:

Additional Editor Comments (if provided):

There some minor points:

1, The Discussion suggests that TEE-guided changes may have mitigated higher intrinsic surgical risk and therefore resulted in similar postoperative outcomes between groups. This is a causal inference that is not supported by the study design, because all patients received intraoperative TEE and no control group exists for comparison. Please rephrase sentences to avoid implying that intraoperative TEE reduced postoperative risk.

2, Although the TTE+TEE group had significantly longer preoperative waiting times, the manuscript states that “disease severity may evolve during the waiting period.” However, the data do not show that waiting time independently contributed to surgical management changes. Please clarify the interpretation by avoiding language that implies a causal relationship (e.g., “waiting time leads to disease progression”). This is important for preventing overinterpretation.

3, Although improved, several sections still contain grammatical issues (like "preoperative TTEalone group"). Please have a final round of careful English editing is recommended to ensure clarity and professionalism.

Reviewers' comments:

Reviewer's Responses to Questions

**Comments to the Author**

Reviewer #1: All comments have been addressed

2. Is the manuscript technically sound, and do the data support the conclusions?

Reviewer #1: Yes

3. Has the statistical analysis been performed appropriately and rigorously?

Reviewer #1: Yes

4. Have the authors made all data underlying the findings in their manuscript fully available?

Reviewer #1: Yes

5. Is the manuscript presented in an intelligible fashion and written in standard English?

Reviewer #1: Yes

Reviewer #1: My reviewer comments have been adequately addressed. .......................................................................

**Do you want your identity to be public for this peer review?** For information about this choice, including consent withdrawal, please see our Privacy Policy

Reviewer #1: No

---

## [Author Response · Author response to Decision Letter 2]

31 Dec 2025

Dear Wei Wu,

We are pleased to submit the revised version of our manuscript, “Impact of Intraoperative Transesophageal Echocardiogram on Changes in Surgical Management among Patients Undergoing Cardiovascular Surgery in Thailand” (Manuscript Number: PONE-D-25-35063R1). We would like to express our sincere gratitude for the constructive feedback and helpful comments provided, which have significantly improved the quality of our work

Editor’s comments:

1, The Discussion suggests that TEE-guided changes may have mitigated higher intrinsic surgical risk and therefore resulted in similar postoperative outcomes between groups. This is a causal inference that is not supported by the study design, because all patients received intraoperative TEE and no control group exists for comparison. Please rephrase sentences to avoid implying that intraoperative TEE reduced postoperative risk.

Response: Thank you for raising this important and subtle point, reflecting the meticulous review provided. To address this issue, we have carefully revised the Discussion to adopt a more conservative tone, focusing on associations rather than causal links. We have also explicitly stated the limitations of interpreting these results due to the observational study design and the absence of a control group for comparison, both in the Discussion and the Limitations section.

Changes in the Manuscript:

From: “TEE-guided interventions likely identified and corrected unanticipated findings in patients with more complex or severe pathology, thereby mitigating higher inherent risks and yielding outcomes similar to those in the less complex “no-change” group.”

To: “Intraoperative TEE facilitated the identification of unanticipated findings in patients with complex or severe pathology. While postoperative outcomes were similar between the 'change' and 'no-change' groups, the observational study design and the universal use of intraoperative TEE preclude a direct assessment of whether these modifications prevented adverse outcomes.”

Added to the final paragraph of the Discussion: “Furthermore, the current study was not designed to establish a direct causal link between these modifications and a reduction in postoperative risk, or it may be that such benefits are not fully reflected in short-term postoperative metrics.”

2, Although the TTE+TEE group had significantly longer preoperative waiting times, the manuscript states that “disease severity may evolve during the waiting period.” However, the data do not show that waiting time independently contributed to surgical management changes. Please clarify the interpretation by avoiding language that implies a causal relationship (e.g., “waiting time leads to disease progression”). This is important for preventing overinterpretation.

Response: We appreciate this guidance to avoid overinterpretation of the relationship between waiting times and clinical evolution. We have revised this section of the Discussion to ensure our conclusions remain strictly aligned with our data. Specifically, we have removed language implying a causal link between waiting time and disease progression and instead report the findings as a lack of independent association in our multivariable analysis.

Changes in the Manuscript:

From: “These findings may suggest that disease severity and case complexity were already well-accounted for in surgical scheduling. Nonetheless, disease status may continue to evolve during the preoperative period, particularly in complex cases. This assumption is supported by the findings in S2 Table, which show no significant differences in waiting times between patients with and without management changes, in both the TTE-only and TTE with TEE subgroups. Still, the waiting time is a risk factor for morbidity and mortality outcomes after cardiac surgery.”

To: “While disease status may potentially evolve during the preoperative period, particularly in complex cases, this data showed no significant difference in waiting times between patients with and without management changes in either the preoperative TTE alone or preoperative TTE combined with TEE subgroups (S2 Table). These findings suggest that, in this study, waiting time was not independently associated with surgical management changes, though it remains a known risk factor for morbidity and mortality after cardiac surgery [19, 20].”

3, Although improved, several sections still contain grammatical issues (like "preoperative TTEalone group"). Please have a final round of careful English editing is recommended to ensure clarity and professionalism.

Response: We sincerely appreciate the Editor’s feedback regarding the manuscript's presentation. To ensure a high standard of clarity and professionalism, the entire manuscript has undergone a final round of meticulous English language editing. We have specifically focused on:

• Terminology Consistency: We have standardized all terms throughout the manuscript. The preoperative groups are now consistently referred to as the "preoperative TTE alone group" and the "preoperative TTE combined with TEE group." The intraoperative assessment is consistently termed "intraoperative TEE."

• Typo error: We have corrected all spacing issues (e.g., "TTEalone") and ensured that mathematical symbols and ranges use the appropriate formatting.

• Grammatical Integrity: The manuscript was reviewed for grammatical accuracy using Gemini (Google) for initial identification of potential errors, followed by extensive manual editing and expert review to ensure technical and clinical precision. While we believe the text is now clear and professionally polished, we remain available for further refinement should any minor stylistic points remain.

• We have conducted a thorough audit of the reference list to ensure accuracy, technical consistency, and full compliance with PLOS ONE guidelines. This process included cross-referencing all citations against retraction databases to confirm that no retracted works are included in the manuscript. Additionally, we carefully evaluated all reviewer-suggested literature, incorporating those publications that provided relevant academic context to enhance our study’s discussion. All references have been verified for correct formatting and are accurately cited within the text.

Reviewer’s comments

All comments have been adequately addressed

We believe these revisions have significantly strengthened the manuscript and hope the revised version now meets the expectations of the Editor and the reviewers. We look forward to your decision and thank you once again for your time and thoughtful consideration of our work.

Sincerely,

Naruenart Lomarat

---

## [Editor Report · Decision Letter 2]

4 Jan 2026

Impact of Intraoperative Transesophageal Echocardiogram on Changes in Surgical Management among Patients Undergoing Cardiovascular Surgery in Thailand

PONE-D-25-35063R2

Dear Dr. Lomarat,

We’re pleased to inform you that your manuscript has been judged scientifically suitable for publication and will be formally accepted for publication once it meets all outstanding technical requirements.

Kind regards,

Wei Wu

Academic Editor

PLOS One

Additional Editor Comments (optional):

The aruthors have fully addressed all concerns.
---

## [Editor Report · Acceptance letter]

PONE-D-25-35063R2

PLOS One

Dear Dr. Lomarat,

I'm pleased to inform you that your manuscript has been deemed suitable for publication in PLOS One. Congratulations! Your manuscript is now being handed over to our production team.

Kind regards,

on behalf of

Dr. Wei Wu

Academic Editor

PLOS One